# Electric Field-Assisted Filling of Sulfonated Polymers in ePTFE Backing Material for Fuel Cell

**DOI:** 10.3390/membranes12100974

**Published:** 2022-10-05

**Authors:** Tung-Li Hsieh, Wen-Hui Guo, Mei-Ying Chang, Wen-Yao Huang, Hsin-Yi Wen

**Affiliations:** 1Department of Electronics Engineering, National Kaohsiung University of Science and Technology, Kaohsiung 80778, Taiwan; 2Department of Photonics, National Sun Yat-sen University, Kaohsiung 80424, Taiwan; 3Department of Chemical and Materials Engineering, National Kaohsiung University of Science and Technology, Kaohsiung 80778, Taiwan

**Keywords:** proton-exchange membrane, electric field filling, composite ePTFE membrane, polyarylene ethers, sulfonated polymer, dimensional stability, swelling ratio, proton conductivity

## Abstract

This study fabricated a composite ePTFE-backed proton-exchange membrane by filling the pores on the ePTFE backing with sulfonated polyarylene ethers through an externally supplied electric field. The morphology changes were observed under an SEM. The results suggested that the application of an electric field had led to the effective filling of pores by polymers. In addition, the composite membrane featured good dimensional stability and swelling ratio, and its water uptake, proton conductivity and component efficiency increased with voltage. It is found in this study that the external application of an electric field resulted in the effective filling of pores in the ePTFE by sulfonated polyarylene ether polymers and, thus, an improved composite membrane performance.

## 1. Introduction

Hazardous climate events in recent years, such as rising sea levels and PM_2.5_ pollution, have increased the public’s environmental awareness. Scientists estimate that fossil fuels will be depleted in 2070 due to the massive need for transportation. To this end, the development of alternative energy sources is on the rise. However, most renewable energy sources are intermittent, opening spatial and temporal gaps between the availability of the energy and its consumption by end-users. Hence, it is necessary to develop suitable energy storage systems for the power grid [1]. Fuel cells combined with water electrolysis can be a promising direction for these purposes. Hydrogen-based fuel cells are a promising source of energy because they produce energy merely through the fuel supplied to them by converting chemical energy into electric energy, with a theoretical conversion of 90%. A proton-exchange membrane fuel cell (PEMFC) consists of a polymer membrane for the electrolyte in the cell [2,3]. The solid polymer membrane effectively reduces the risk of electrolyte infiltration to other layers. Some famous examples of widely-used commercialized PEMFC materials are Dupont’s Nafion membranes, Asahi Glass’ Aciplex and Dow Chemical’s Dow perfluorosulfonated solid-state electrolytes [4,5,6]. The advantages of perfluorosulfonated polymers have been reported, such as hydrophobic PTFE backbone, hydrophilic sulfonic acid groups, phase separation when hydrated, excellent proton conductivity and so on [7,8].

As the core material in a fuel cell, a proton-exchange membrane (PEM) works by transmitting H^+^ generated at the anode to the cathode for reaction, as well as isolating gases at both the anode and cathode to avoid direct contact and explosion risks. Therefore, it requires a very low gas penetration rate. It also requires a certain level of mechanical strength to adapt to environments with high temperatures and humidity. High proton conductivity is required for proton transmission. A PEM is considered excellent if it has the aforementioned features. Despite the outstanding performance of Nafion as a PEM [9,10], the reduced mechanical strength under high temperatures and the increased difficulty of development and production due to its perfluorinated structure have led to excessive production costs and high retail prices. Additionally, the fluorinated precursors during the production could potentially contaminate the environment. These factors have prompted many research teams to search for an alternative solid-state polymer electrolytes, such as sulfonated polymer derivatives such as sulfonated polyetheretherketone (SPEEK), sulfonated polysulfone (SPSF), sulfonated polysulfideulfone (SPSS) and polybenzimidazole (PBI) [11,12], or the improvement of Nafion with inorganic additives [13,14,15,16]. In addition, pore filling has been extensively adopted to improve the stability of the Nafion membrane. At the Wuhan University of Technology, Haolin Tang and his team soaked Nafion in a solution of chemically modified, expanded polytetrafluoroethylene (ePTFE) [17]. The fuel cell stability was expected to be improved by filling up the backing material. Membrane modification through an electric field is also a unique way of improving efficiency. Lin et al. were the first to apply an electric field on Nafion as a solid polymer, and they found that the proton conductivity in the membrane had improved [18]. In addition to Nafion, an electric field has been applied to materials such as sulfonated polyphenyleneoxide (PPO), and improvement has been observed in terms of membrane proton conductivity [19,20,21]. The material series previously developed by our team is a novel polyarylene ether polymer with a high glass transition temperature [22,23]. Based on previous studies, A-series material features fairly good thermal stability. The sulfonation fluorinated poly(aryl ether)s (SA8) (as Figure 1) with the linkage of ether (–O–) between fluorinated monomers and bisphenols that was reported by Chun-Che Lee [24] has a proton conductivity twice greater than that of Nafion 117; the hydrogen fuel cell element made of SA8 yields good efficiency. However, the series of polymers are prone to excessive water uptake in an environment of high temperature and humidity due to the sulfonation, and the membrane swells or even is dissolved in water; moreover, the material has a poor service life, as the catalyst layer easily peels off due to the excessive swelling ratio during component efficiency measurement.

The SA8 polymer with good overall performance was selected for this study. The membrane-forming technique proposed in the literature [25] was introduced, and an SA8 solution was used to fill the ePTFE pores, while a vertical electric field was applied externally to generate an SA8 membrane. The ePTFE is a porous membrane created by expanding and stretching polytetrafluoroethylene. It features good thermal stability and wide application temperatures. Its excellent chemical stability helps stabilize the membrane’s changes in size. As a result, the addition of ePTFE backing is expected to improve the mechanical performance of the membrane, provide the SA8/ePTFE composite membrane with good dimensional stability and stable swelling ratio in a highly sulfonated environment and maintain the good water uptake of SA8 and the electric properties of the membrane.

## 2. Materials and Methods

### 2.1. SA8 Procedure for Sulfonation

The SA8 polymerization reactions were performed using standard vacuum-line techniques and dehydrated with a slow stream of nitrogen in the reaction vessel throughout the reaction. Ref. [25] Poly(arylene ether)s synthesized using a multiphenylated difluoro monomer with a trifluoromethylphenyl side chain and three types of multiphenylated bisphenol monomer, which become partially fluorinated sulfonated poly(arylene ether)s with highly efficient proton transport [26].

To a solution of the polymer SA8 [24] (1.20 g) in DCM (75 mL) at room temperature, chlorosulfuric acid in dichloromethane (DCM) was added dropwise. The reaction mixture was stirred for 24 h and then poured into water. The SA8 precipitate was filtered, washed thoroughly with DI water until the pH became neutral and then dried in a vacuum at 60 °C overnight to obtain the sulfonated polymer. The polymers were sulfonated to different extents by using the aforementioned procedure and adding 4 mL of a sulfonating agent, respectively. The sulfonated polymer was readily soluble in polar aprotic solvents such as dimethylformamide, dimethylacetamide (DMAC), dimethyl sulfoxide (DMSO) and n-methyl-2-pyrrolidone (NMP). The series of materials developed in the laboratory in the past polyarylether polymers with a high glass transition temperature. From the previous research by C.C. Lee [24], it is known that the thermal stability of a series of materials is quite good. The polymer’s thermal decomposition temperature is as high as 560 °C.

### 2.2. Preparation of Polymer Membrane

The solute was dissolved in the solvent using phase transition [27], and the solution was made into a gel state. The sulfonated polymers were placed in a vial, and DMSO was added. A small magnet was added to the vial, which was then rotated and stirred in a sand bath at 80 °C on a hot plate, improving the dissolution of sulfonated polymers in DMSO. As all the polymers were dissolved, the solution was de-aerated, and larger particles and debris were filtered out. The filtered sulfonated polymers were transferred to a medium-sized vial. An ePTFE (Hydrophilic ePTFE Membrane, Valueplus Materials Inc., Taiwan) was cut to the appropriate size. This membrane was soaked in the filtered sulfonated polymer solution. With the membrane in it, the solution was placed on a piece of ITO conductive glass with the cathode electrode fixed on it [25]. As Figure 2, the anode electrode was fixed on the membrane coating bar wrapped on an iron piece. DC voltages of 0, 1, 3 and 5 V were injected, and an automatic membrane coater (ZAA 2300, ZEHNTNER) was used to create a membrane at 0.3 mm/sec. At the end of the coating, the sulfonated polymer composite membrane was transferred to an oven to bake at 80 °C for at least 12 h to dry the solution and stabilize the membrane shape.

### 2.3. Observation under SEM

The cross-sectional specimen was prepared by cutting the membrane into a 2 × 2 cm square and dipping it in liquid nitrogen. Metal clamps were used to clamp both sides of the membrane to break it and flatten it on a piece of filter paper. The purpose of the nitrogen dipping was to make the membrane brittle and prevent the breaking from compromising the membrane’s shape. Then, the membrane was subject to vacuum evaporation to deposit a thin layer of gold on the surface of the specimen for better electric conductivity. Finally, the specimen was observed under a scanning electron microscope (SEM, KCT Phenom Pro).

### 2.4. X-ray Diffractometer

The membrane was cut roughly to slightly larger than 1.5 × 1.5 cm. Once cut, the membrane was placed in the chamber of an X-ray diffractometer (XRD, Rigaku Miniflex Benchtop X-ray Diffractometer) for measurement. The XRD spectrum was generated with the help of a computer program before the data was subject to qualitative and quantitative analyses.

### 2.5. IEC Measurement

First, at least 0.01 g of the membrane was cut off, and the membrane was dehydrated in a vacuum oven at 80 °C for 3–4 h before the weight of the dry membrane (*W_dry_*, *g*) was measured. Then, the piece of the membrane was soaked in 1 M NaCl _(aq.)_ for at least 24 h, where Na ions replaced the H^+^ in the sulfonates to form SO_3_Na. The H^+^ was released into the solution to make the solution acidic, from which the number of sulfonates was determined. The membrane was removed before measurement, and three drops of phenolphthalein were added as an indicator. A total of 0.01 N NaOH_(aq.)_ was formulated for acid–base neutralization and titration to obtain *V_NaOH_*, which was substituted into the following equation to determine the *IEC* value [28].
$$IEC\left(\mathrm{mmol}/g\right)=\frac{\left(V_{NaOH},\;\mathrm{mL}\right)\times \left(N_{NaOH},\;N\right)}{\left(W_{dry},g\right)}$$

*V_NaOH_*: volume of NaOH (mL)

*N_NaOH_*: equivalent concentration of OH^-^ in NaOH (determined in KHP titration)

*W_dry_*: dry membrane weight (g)

### 2.6. Water Uptake and Dimensional Stability

Water uptake and dimensional stability were measured by placing several pieces of membrane, cut to 2 × 1 cm, in a vacuum oven to bake at 70 °C for 4 h. The dry membrane weight (*W_dry_*) and the thickness of the dry membrane (*L_dry_*) were measured before the membranes were soaked in DI water. Membranes were soaked for 24 h individually under various temperature and humidity conditions. The wet membranes were taken out and dried with a piece of filter paper. The changes in weight (*W_wet_* − *W_dry_*) of the wet membranes were measured and substituted into the equation below to determine data [29].
$$Water\;uptake\left(\Delta wt\%\right)=\frac{W_{wet}-W_{dry}}{W_{dry}}\times 100\%$$
$$Dimensional\;stability\left(\Delta L\%\right)=\frac{L_{wet}-L_{dry}}{L_{dry}}\times 100\%$$

### 2.7. Swelling Ratio Test

The swelling is a phenomenon where the volume of polymers expands between the molecules as they are in solution. The solvent molecules spread in the structure of the polymers, resulting in the expansion of polymer volumes. By comparing the area of the wet membrane (*S_wet_*) and that of the dry membrane (*S_dry_*), the swelling ratio of the material is obtained, which is the difference in the shrinking or swelling of the membrane in dry conditions or in solution, respectively. The swelling ratio is related to the temperature and pressure during the experiment and the-cross linking between polymer molecules and has something to do with the polarity of solute and the solvent it is dissolved in [30].
$$Swelling\;Ratio\left(\Delta S\%\right)=\frac{S_{wet}-S_{dry}}{S_{dry}}\times 100$$

### 2.8. Hydration Number (λ)

Physically, the hydration number (*λ*) indicates how many water molecules are in every unit of sulfonates. It is the ratio of the number of water molecules over that of sulfonates in the membrane. The greater the hydration number, the more water molecules are around the sulfonate. Protons are easily transferred in PEM if the *λ* value is appropriate. By substituting the known *IEC* value and the water uptakes at every temperature into the equation below, the value of *λ* was determined [31,32].
$$\lambda =\frac{\left[H_{2}O\right]}{\left[SO_{3}{}^{-}\right]}=\frac{(W_{wet}-W_{dry})\times 1000}{W_{dry}\times M_{H2o}\times IEC\left(\mathrm{mmol}/g\right)}=\frac{Water\;uptake\times 10}{M_{H2o}\times IEC\left(\mathrm{mmol}/g\right)}$$

### 2.9. Proton Conductivity Measurement

The measurement was conducted by cutting the membrane into 1 × 0.5 cm (L × W) pieces, measuring their thickness and placing them in a constant temperature and humidity chamber (ESPEC SH-241). The measurement was performed at a constant temperature of 80 °C and various humidity conditions. Once the temperature was stabilized, and the relative humidity reached the set value, the pieces to be measured were connected to an AC impedance analyzer (Agilent 4294A) with clamps (16089B). At this moment, the voltage modulation parameter was 10 mV for the analyzer, and the range of scan spectrum was 10–10 MHz. The impedance analyzer was regulated to generate an impedance value that was substituted into the following equation to determine the proton conductivity of the membranes measured.
$$\sigma \left(s/\mathrm{cm}\right)=\frac{L\left(\mathrm{cm}\right)}{R\left(\Omega \right)A\left({\mathrm{cm}}^{2}\right)}$$
where R(Ω) is the measured impedance;

$A\left({\mathrm{cm}}^{2}\right)$ is the membrane’s cross-sectional area (membrane thickness × width); and

L(cm) is the thickness of the membrane.

### 2.10. Membrane Electrode Preparation and Single Cell Efficiency Measurement

The catalyst slurry was made of Pt/C (0.217 g, Platinum 40% on carbon black, Alfa Aesar), commercial Nafion D520 dispersion (1.743 g, 5%wt), methanol (23.009 g) and DI water (5.948 g). The material was placed in a vial that was shaken in an ultrasonic bath to ensure even distribution. The catalyst slurry was then poured into the feeder of an ultrasonic atomizer sprayer (PRISM-400 BT Benchtop) to spray the slurry evenly over a 3 × 3 cm PEM at a constant sprayer chamber temperature of 60 °C. Once sprayed, the PEM was made into 1 × 1 cm membrane electrodes along with a gas diffusion layer (0.235 mm, GDL, Carbon paper Sigracet 28BC) and a Teflon pad (0.151 mm). The Pt loading was 0.4 and 0.2 mg/cm^2^ for the cathode and anode, respectively. A single cell was exposed to pure hydrogen (0.4 L/min) and pure oxygen (0.2 L/min) with the intake temperature maintained at 80 °C and relative humidity at 100% RH for both gas flows.

## 3. Results

### 3.1. SEM Microstructure Analysis

As shown in Figure 3, the pure ePTFE membrane was very porous and had a net-like structure. In Figure 4a, it is clear that SA8 had formed a dense polymer coating on both sides of ePTFE without the application of an electric field, and there was a distinct boundary between SA8 and ePTFE. By increasing the voltage to 1 V, it was found that the ePTFE pores were filled by SA8; however, the distinct boundary between SA8 and ePTFE still existed, as shown in Figure 4b. In Figure 4c, the voltage was increased to 3 V, and more ePTFE pores were efficiently filled by SA8, which gradually eliminated the boundary between SA8 and ePTFE. At 5 V, as shown in Figure 4d, it is clear that not much SA8 was left on either side of ePTFE, suggesting that almost all SA8 was spent in filling the ePTFE pores. It is also found in the composite membrane that the pores of pure ePTFE were almost enclosed by SA8. Therefore, it is safe to say that the external application of voltage enabled the effective filling of sulfonated polymers in ePTFE backing, and these pores were better filled by more polymers as the voltage increased.

### 3.2. XRD Diffraction Diagram Analysis

For the same matter, the peak value area indicates the content of crystals when tested with the same instrument under the same test conditions; i.e., the greater the area, the more crystals; the narrower the peak, the bigger the crystal grains; and the higher the peak, the better the crystallization. It is seen in Figure 5 and Figure 6 that the peak strength of the ePTFE membrane increased with voltage, proving that the application of voltage facilitated the improvement of crystallization in SA8 polymers.

### 3.3. IEC Value of Composite Membrane

The IEC values of the membrane are provided in Table 1. The data indicates that the membrane IEC increased with voltage, suggesting that more SA8 polymer molecules found their way into ePTFE pores, thus improving the ion exchange capability of the membrane.

### 3.4. Water Uptake and λ Value of Membrane

Figure 7 presents water uptake of the membrane vs. temperature. The water uptake maintained a stable range despite the temperature rise. At 90 °C, the water uptake of the ePTFE membrane was better than that of Nafion 211 and was maintained below 100%. Clearly, the addition of ePTFE had no effect on the high-water uptake of SA8. The water uptake increased with voltage, as ePTFE-5 V had the highest water uptake (93.75%), suggesting that the increased electric field had allowed more SA8 polymers to penetrate into ePTFE pores and increased the water uptake.

The λ value (nH_2_O/nSO_3_H) represents the number of water molecules in every unit of sulfonate. Protons are easily transferred in PEM at a suitable λ value. In the experiments, the calculated λ values are listed in Table 2. The data indicates that the λ values of ePTFE were not very different from those of Nafion 211. Therefore, it is expected that the proton conductivity of the ePTFE membrane should produce satisfactory results.

### 3.5. Dimensional Stability of Membrane

Figure 8 shows the changes in the stability of ePTFE membrane thickness with temperature, while Figure 9 shows the changes in the stability of ePTFE membrane width with temperature. Both results were compared with the commercial Nafion 211 membrane. The width changes were larger since the scale was smaller than those of the thickness. However, the same trend was still observed in the thickness stability. All membranes remained intact at 90 °C. Based on the data, the pure SA8 membranes had experienced greater dimensional changes (30.79% and 33.44%) than Nafion 211 (13.95% and 12.45%). However, the addition of ePTFE backing significantly improved the dimensional stability, as the dimensional changes of ePTFE remained between 10% and 14% at 90 °C. Based on these changes, the dimensional stability of ePTFE membranes exhibited no significant changes despite the increasing temperature, and the changes all fell within a 5% tolerance range. Therefore, the membranes with backing had good dimensional stability even with the application of various voltages, verifying that the addition of backing improved the dimensional stability of the membranes effectively.

### 3.6. Swelling Ratio of Membrane

The changes in the swelling ratio of SA8 and ePTFE membranes with temperature are provided in Figure 10. The results were compared with those of the commercial Nafion 211 membrane as well. The swelling ratio is the ratio of volume expansion between molecules as polymers are dissolved in a solvent. The data indicates that the swelling ratio of the pure SA8 membrane was 80% at high temperatures; with the backing of ePTFE, however, the ratio fell to between 20% and 25% for all ePTFE membranes, not far from that of Nafion 211 (28%). Further, the swelling ratio of ePTFE membranes did not experience significant changes with temperature. Clearly, the backing improved the poor swelling ratio in SA8, and the ratio remained satisfactory at high temperatures.

### 3.7. Proton Conductivity Data Analysis for Composite Membranes

The impedances of pure SA8 membranes and ePTFE membranes measured using the AC Impedance at a constant temperature of 80 °C and varying relative humidity were substituted into the equation to yield the proton conductivity, as shown in Figure 11, and the results were compared with those of the commercial Nafion 211 membrane. In an environment with low humidity, the proton conductivity was between 15.68 and 20.99 mS/cm for ePTFE membranes, not far from those of pure SA8 membranes and Nafion 211 membranes. When the humidity was increased to 95%, however, the proton conductivity came to 124.21 mS/cm for ePTFE-5 V and 152 mS/cm for ePTFE-3 V, which were both higher than the 118.2 mS/cm of Nafion 211.

According to the literature [9,10,11,17,25,26,27], the ion channel is much more in order than that of the original membrane due to the process of electron alignment, and the ion channel in the membrane becomes shorter, which leads to the effective improvement of proton conductivity of the membrane. Therefore, in theory, the conductivity of the membrane should increase with the electric field. However, it is found in Figure 11 that the proton conductivity was higher in ePTFE-3 V than in ePTFE-5 V. Despite, on average, that the data of ePTFE-3 V appeared better, the range of error was larger for ePTFE-3 V with the error bars plotted, and the ePTFE-5 V performed with greater stability, as shown in Figure 12. Therefore, it can be posited that the increase in voltage helped improve the proton conductivity of the membrane.

### 3.8. Fuel Cell Component Efficiency Analysis

The efficiency of fuel cell components measured for the composite membranes at a constant temperature of 80 °C is provided in Figure 13 and Figure 14. The component efficiency plots are discussed in two parts for the sake of data analysis.

The ePTFE membrane fuel cell component efficiency is plotted in Figure 13. For ePTFE-0 V, ePTFE-1 V, ePTFE-3 V and ePTFE-5 V, the open-circuit voltage was 0.848, 0.847, 0.848 and 0.847 V; the maximum current density was 2490, 2607, 2859 and 2898 mA/cm^2^; and the maximum component efficiency was 0.8, 0.83, 0.89 and 0.94 W/cm^2^, respectively.

The experiment results revealed that the component efficiency increased for the composite membranes with the voltage. It could be because the water uptake and λ value had increased with the voltage. Water is the carrier for proton transmission. Therefore, the ePTFE membranes performed similarly in terms of dimensional stability and swelling ratio and exhibited no swelling or rupture during the component efficiency measurement, but ePTFE-5 V displayed better proton conductivity thanks to its better water uptake and λ value and had a relatively stable proton conductivity, which could explain why its component efficiency was better than those of the other ePTFE membranes and why the component efficiency increased with the electric field.

ePTFE-5 V, the most efficient of the ePTFE membranes, was selected for a comparison with the commercial Nafion 211 membrane and SA8 membrane, as shown in Figure 14. For SA8, ePTFE-5 V and Nafion 211, the open-circuit voltages were 0.747, 0.847 and 0.797 V; the maximum current densities were was 2477, 2898 and 2861 mA/cm^2^; and the highest component efficiencies were 0.91, 0.94 and 0.93 W/cm^2^; respectively.

The experimental results indicated that the component efficiency of ePTFE-5 V was slightly better than those of SA8 and Nafion 211. The water uptake and proton conductivity were lower in ePTFE-5 V than in SA8, but the dimensional stability and swelling ratio were better than in SA8, given that ePTFE-5 V had better membrane stability during component measurement, which is probably why the component efficiency was slightly better than SA8. With this in mind, the addition of ePTFE backing could improve the dimensional stability of PEM effectively in an environment of high temperature and humidity. The forming of the membrane by applying voltage allowed the water uptake and proton conductivity to stay above standard without compromising the membrane’s electric conductivity, thus making the component efficiency slightly better than SA8.

## 4. Conclusions

A proton-exchange membrane containing sulfonated polyarylene ether polymers and ePTFE backing was successfully produced by applying an electric field. The changes in membrane properties after the application of the electric field were investigated in addition to the filling of pores in sulfonated polyarylene ether polymers. The results revealed optimized advantages in IEC value, water uptake, dimensional stability, swelling ratio, proton conductivity and component efficiency. Particularly, the performance was outstanding for dimensional stability and swelling ratio, as both experienced no significant changes with the temperature, thus improving the less than satisfactory dimensional stability and swelling ratio in SA8.

## 5. Patents

Patents resulting from the work reported in this manuscript: US8,987,407B2; US9,748,594B2; US9,644,069B2; US9,209,472B2; US9,018,336B2.

## Figures and Tables

**Figure 1 membranes-12-00974-f001:**
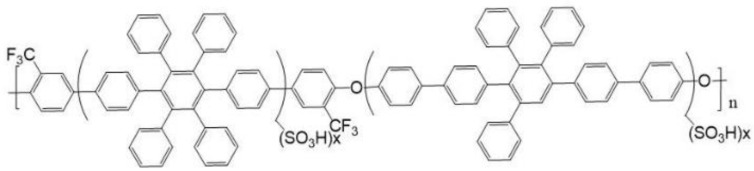
Sulfonated polymer structure of SA8 [24].

**Figure 2 membranes-12-00974-f002:**
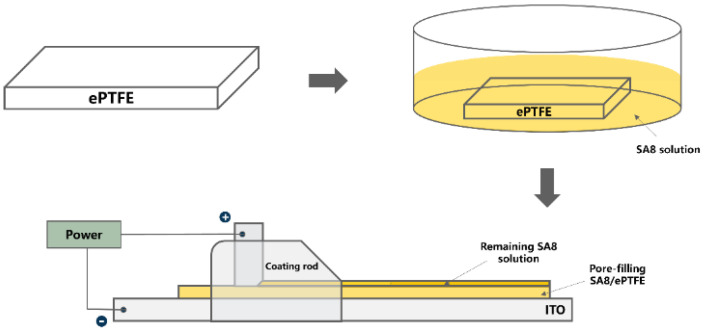
Production of the polymer membrane.

**Figure 3 membranes-12-00974-f003:**
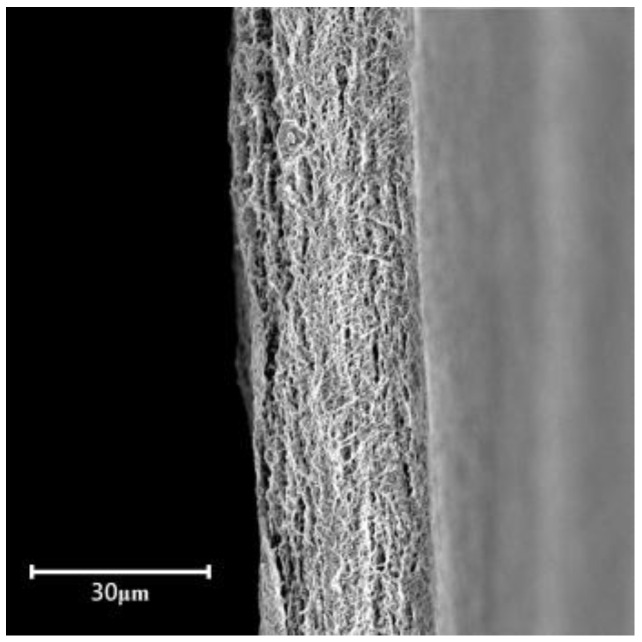
Cross-section of pure ePTFE membrane.

**Figure 4 membranes-12-00974-f004:**
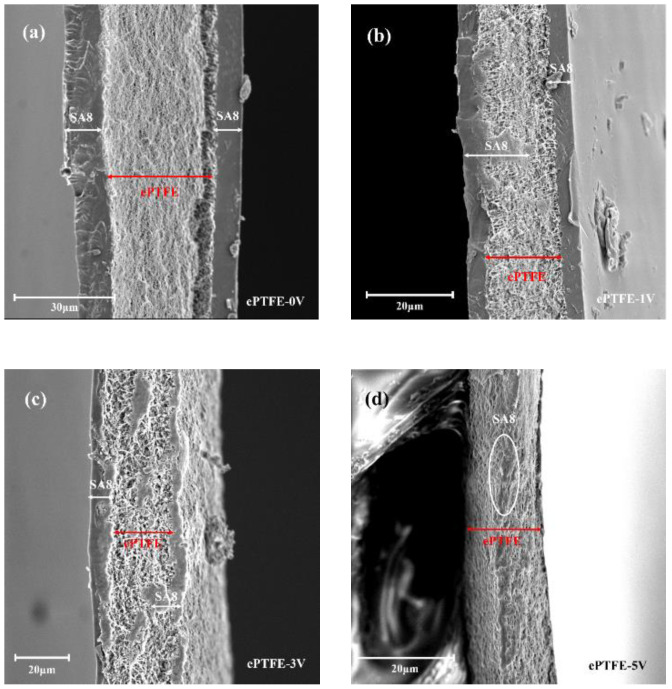
Cross-sections of ePTFE membranes under SEM: (**a**) ePTFE-0 V at 1:2800 scale; (**b**) ePTFE-1 V at 1:3600 scale; (**c**) ePTFE-3 V at 1:4400 scale; (**d**) ePTFE-5 V at 1:4100 scale.

**Figure 5 membranes-12-00974-f005:**
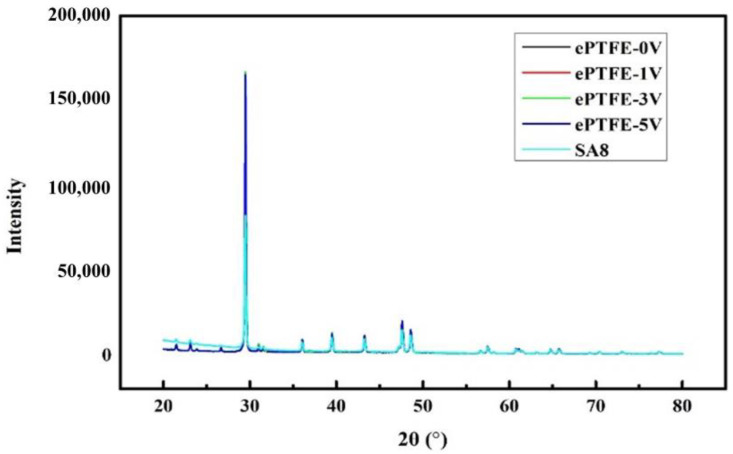
XRD diffraction section of SA8 and ePTFE membrane, 20–80°.

**Figure 6 membranes-12-00974-f006:**
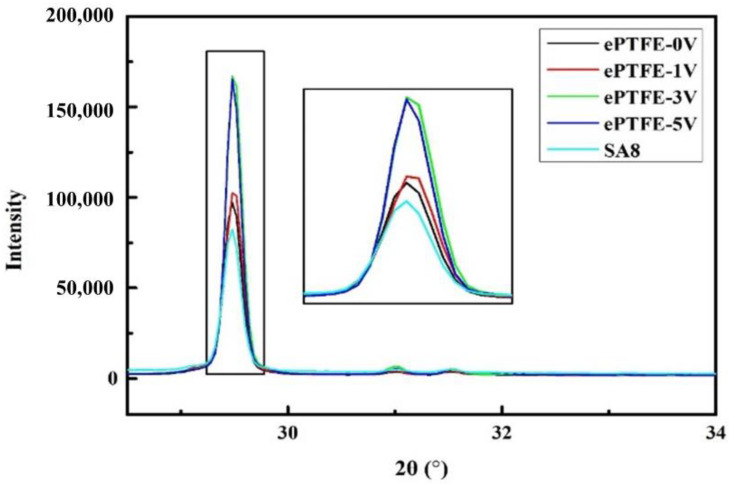
XRD diffraction section of SA8 and ePTFE membrane, 28–34°.

**Figure 7 membranes-12-00974-f007:**
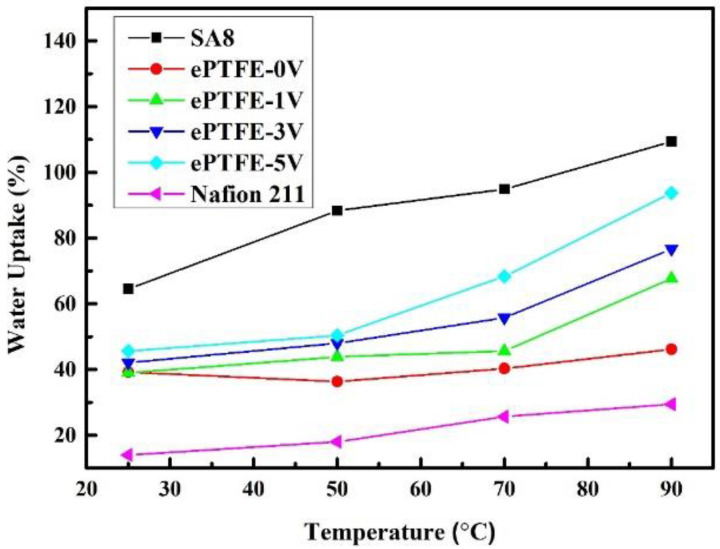
Water uptake of ePTFE membranes at various temperatures.

**Figure 8 membranes-12-00974-f008:**
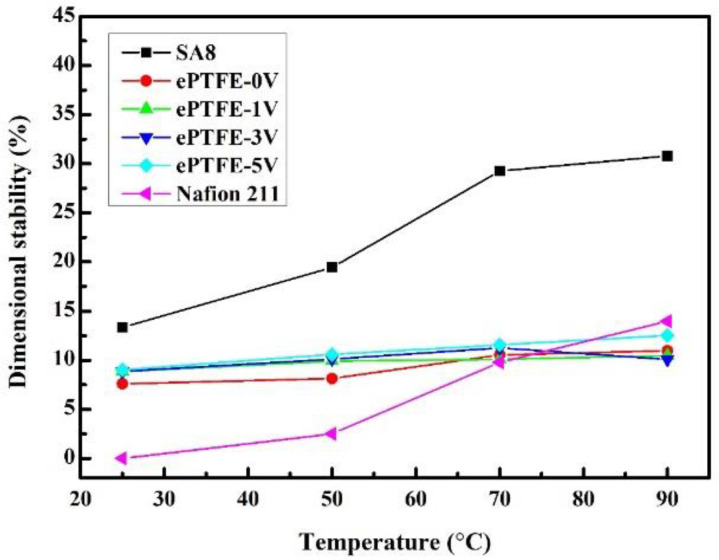
The thickness stability of ePTFE membranes at various temperatures.

**Figure 9 membranes-12-00974-f009:**
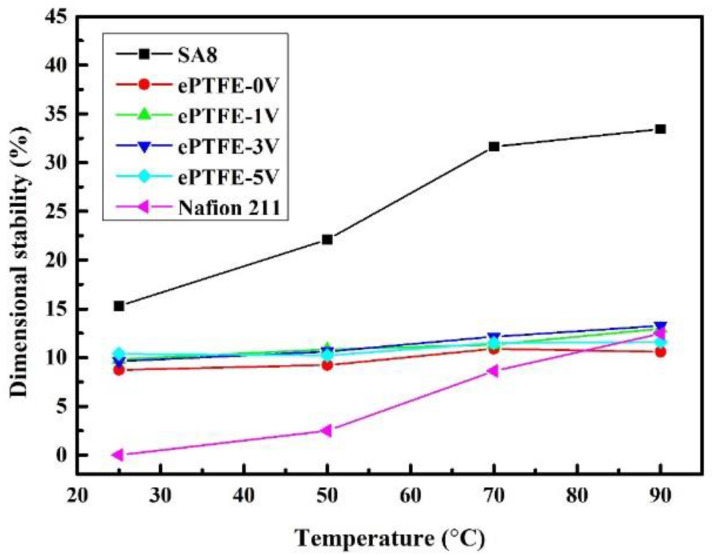
Width stability of ePTFE membranes at various temperatures.

**Figure 10 membranes-12-00974-f010:**
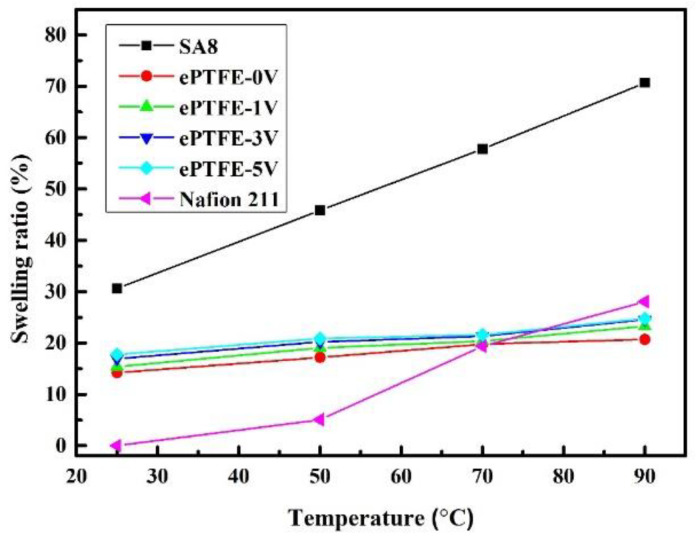
Swelling ratios of ePTFE membranes at various temperatures.

**Figure 11 membranes-12-00974-f011:**
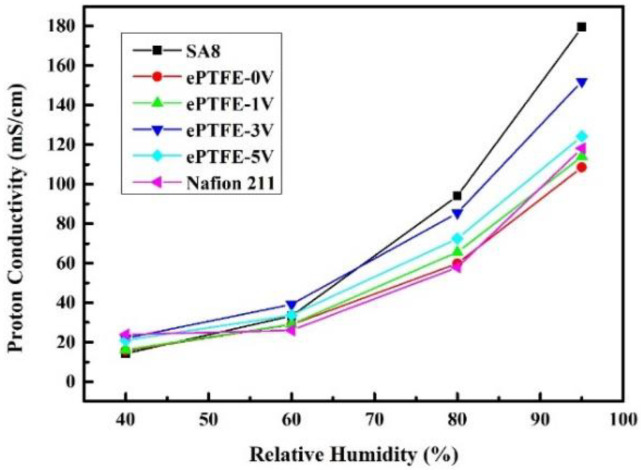
Proton conductivity of ePTFE membranes at 80 ℃ and various relative humidities.

**Figure 12 membranes-12-00974-f012:**
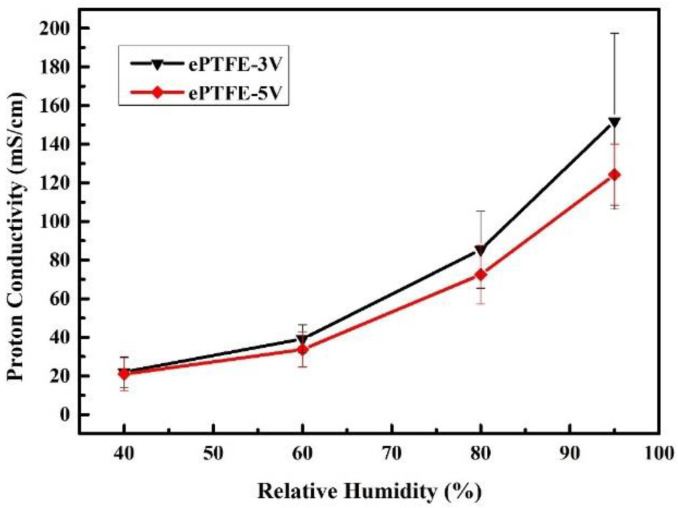
Proton conductivity of ePTFE-3 V vs. ePTFE-5 V at 80 °C and various relative humidities.

**Figure 13 membranes-12-00974-f013:**
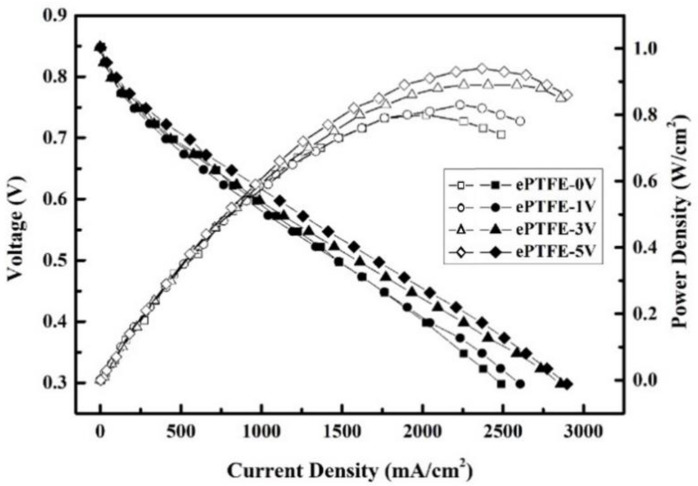
Fuel cell component efficiency of ePTFE membranes at the intake temperature of 80 °C and relative humidity of 100%RH.

**Figure 14 membranes-12-00974-f014:**
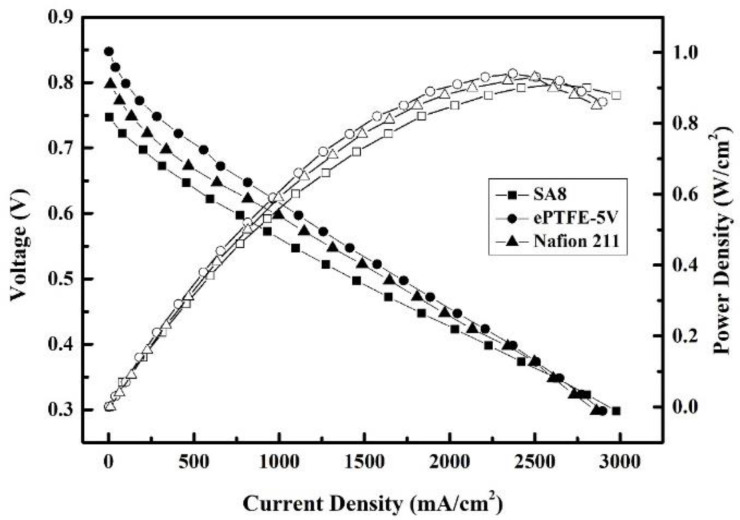
Fuel cell component efficiency of SA8, ePTFE-5 V and Nafion 211 at an intake temperature of 80 °C and relative humidity of 100%RH.

**Table 1 membranes-12-00974-t001:** IEC values of SA8 and ePTFE membranes.

Sample Name	IEC (Tirt.) (mmol/g) ^a^	IEC (Theo.) (mmol/g) ^b^	Degree of Sulfonation (%) ^c^
**SA8**	3.01	3.71	81.20
**EPTFE-0 V**	1.95	1.34	145.24
**EPTFE-1 V**	1.98	1.89	104.98
**EPTFE-3 V**	2.1	2.13	98.66
**EPTFE-5 V**	2.18	2.32	93.84

^a^ IEC (tirt.): IEC value obtained by titration; ^b^ IEC (theo.): theoretical IEC value for full replacement. ^c^ IEC from titration/IEC from maximum replacement of sulfonate: DS (%) = IEC (titr.)/IEC (theo.) × 100.

**Table 2 membranes-12-00974-t002:** λ values of SA8 and ePTFE membranes.

Sample	Hydration Number (Λ)			
	25 °C	50 °C	70 °C	90 °C
**SA8**	11.91	16.30	17.51	20.20
**EPTFE-0 V**	11.19	10.36	11.49	13.16
**EPTFE-1 V**	10.95	12.32	12.80	18.99
**EPTFE-3 V**	11.13	12.70	14.75	20.29
**EPTFE-5 V**	11.62	12.85	17.42	23.89
**NAFION 211**	8.55	10.99	15.68	18.00

## Data Availability

Not applicable.

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
