# Peer review of "Electric Field-Assisted Filling of Sulfonated Polymers in ePTFE Backing Material for Fuel Cell"

_membranes, 2022, doi:10.3390/membranes12100974_

Round 1

Reviewer 1 Report

The manuscript deals with ePTFE membrane filled with sulfonated polymer using an electrical field 

The manuscript structure is comprehensive and membranes are very well characterized 

The manuscript could be accepted after addressing the following points

1- In the introduction section -SA8 should be defined in detail.

2-Material and Methods parts are repeated with the same section number as the introduction. There is also a meaningless paragraph from line 83to line 91

The 3-sulfonation process should be described in detail.

4- TGA and Mechanical tests measurements are missing, which are necessary to check the thermal stability and mechanical strength of the membranes.

5- Cross-section SEM figures need to be rotated for better visibility.

6- The naming of the membranes should be mentioned in the material and methods section.

7- In table 7, what is the aim of measuring the degree of sulfonation after preparing the membranes, what is this DS refer to?? Is it the sulfonation degree before or after the membrane preparation??

Author Response

Dear Oliver Fan

I am writing to you regarding our paper entitled” Electric Field-Assisted Filling of Sulfonated Polymers in ePTFE Backing Material for Fuel Cell”, Manuscript ID: membranes-1950504.

We greatly appreciate your comments and those of the reviewer. The comments have helped us to improve our manuscript considerably. The following is a list of the original comments along with our responses, which also specify how and where the manuscript was modified. Changes made in the manuscript are marked in a “red” font. The revisions to the manuscript were made in consultation with all of the coauthors, and each author has given their approval to the final form of the revised manuscript revision.

We hope that the revised manuscript will be deemed suitable for publication. We thank you and the reviewer for the consideration thus far, and we look forward to hearing from you again soon.

Sincerely,

Hsin-Yi Wen

Assistant Professor,

Department of Chemical and Materials Engineering,

National Kaohsiung University of Science and Technology,

TEL: 886-7-3814526 ext 15110,

FAX: 886-7-3830674,

Postal address: No. 415, Jiangong Rd., Sanmin Dist., Kaohsiung City 80778, Taiwan

Reviewer 2 Report

The manuscript reported the fabrication of a composite ePTFE-backed proton exchange membrane by filling the pores with sulfonated polyarylene ethers through an externally supplied electric field. The electric field may lead to the effective filling of pores by polymers. The obtained composite membrane demonstrated good dimensional stability and swelling ratio, and its water uptake, proton conductivity and component efficiency increased with voltage.

I consider the content of this manuscript will meet the reading interests of the readers of the Membranes journal. However, there are certain English spelling and grammar issues, and also the discussion and explanation should be further improved. I suggest giving a minor revision and the authors need to clarify some issues or supply some more experimental data to enrich the content. This could be comprehensive and meaningful work after revision.

Detailed comments can be found in the PDF file.

Author Response

(The authors gave the same response as above.)
